# OpenReview forum: "ASRU: Activation Steering Meets Reinforcement Unlearning for Multimodal Large Language Models"
_ICML.cc/2026/Conference — ICML 2026 regular_

### Official Review · Reviewer_wYV4 · 2026-03-11

**Soundness:** 3
**Presentation:** 3
**Significance:** 3
**Originality:** 2
**Overall Recommendation:** 4
**Confidence:** 3

**Summary:**

This paper addresses a limitation of existing multimodal unlearning methods: that they optimize solely for output deviation from ground truth while neglecting the naturalness of post-unlearning responses, often producing hallucinated or rigid refusals. Overall, the work examines a issue in practical MLLM deployment—that unlearned models must behave coherently rather than merely suppress target outputs. This paper finds that generation quality should be treated as a primary evaluation criterion alongside forgetting efficacy. To this end, the paper proposes ASRU, a two-stage framework that first induces refusal behavior via activation steering on a single down-projection layer, then refines a fine-grained forget–retain boundary using GRPO with a customized reward function and a boundary set drawn from the retain set.

**Compliance With Llm Reviewing Policy:**

Affirmed.

**Final Justification:**

My concerns have been solved.

**Key Questions For Authors:**

- The generation quality evaluation would benefit substantially from a human validation study, even on a small subset, to confirm that GPT-4o Mini scores correlate with human preference. Reporting the prompt sensitivity of the rubric (e.g., rephrasing the scoring criteria) would also help establish metric reliability.

- It would strengthen the paper to include an analysis of the steering vector vL* itself—for example, showing that projecting out vL* from forget-set activations reduces forget-specific information while leaving retain-set activations largely unchanged, to support the claim that the direction is semantically meaningful rather than a generic domain-shift vector.

- Evaluating ASRU on at least one additional model family (e.g., LLaVA or InternVL) or a different unlearning domain would more concretely support the generalization claims made in Section 4.2.

**Limitations:**

yes

**Strengths And Weaknesses:**

**Summary of Strengths:**

- Practically motivated problem framing. Identifying generation quality degradation as a distinct failure mode of existing unlearning methods is a meaningful contribution. The distinction between hallucinated outputs, rigid template refusals, and context-aware dynamic refusals is well-articulated and supported by illustrative examples in Figure 1.

- Efficient two-stage design. Updating only a single down-projection matrix in Stage 1 is computationally lightweight, and the closed-form solution (Eq. 10) provides a principled initialization for the subsequent GRPO stage. The ablation study (Table 2) convincingly demonstrates that both stages are individually necessary.

- Boundary set construction is a useful contribution. Explicitly constructing a boundary set D̃r of retain samples semantically close to the forget set, and incorporating it into the GRPO reward, is a well-motivated strategy for preventing over-refusal and improving boundary discrimination.

**Summary of Weaknesses:**

- The generation quality metric relies heavily on GPT-4o Mini, without validated reliability. Contextual Refusal (CR) and Forgetfulness scores are computed by prompting GPT-4o Mini with a fixed rubric. The paper does not report inter-annotator agreement, correlation with human judgments, or sensitivity to prompt phrasing. Given that ASRU's primary claimed advantage is a 5.8× improvement in generation quality, the validity of this metric is critical, and its current form is insufficiently validated to support quantitative comparison at this scale.

- The knowledge-absence direction construction conflates two distinct concepts. The steering vector vL* is defined as the difference between mean activations on Dtarget (images unseen during pretraining) and Df (the forget set). However, "absence of knowledge" and "absence of pretraining exposure" are not equivalent: a model may have no stored knowledge about a forget-set identity while still producing similar activations to Dtarget for unrelated visual reasons. The paper does not verify that vL* reliably captures a forget-specific direction rather than a domain-shift artifact, nor does it examine whether the direction generalizes across diverse forget-set compositions.

- Experimental scope is narrow, limiting generalizability claims. All main results are reported on a single benchmark (MLLMU-Bench) using Qwen3-VL variants, with a single intervention layer fixed at L*=17 across experiments. The paper claims strong generalization but does not evaluate on other datasets, model families, or knowledge types beyond fictitious personal profiles. Additionally, the guidance strength λ differs substantially between the 4B (λ=3) and 8B (λ=0.2) models without a principled explanation, raising questions about the robustness of the hyperparameter selection procedure.

---

> ### Author Rebuttal · Authors · 2026-03-31
>
> We appreciate your insightful review and address your concerns below.
> >W1 & Q1: Reliability Validation of GPT-4o-mini
>
> **R1:** We further re-evaluate the responses using GPT-5.1, Claude 4.5 and 3 expert annotators. As shown in Table 1, although the scores vary across judges, ASRU still achieves relatively high scores. Based on the results in Table 1 from the response to reviewer dYwq, the relative ranking of methods remains consistent, and ASRU achieves the best performance under all evaluators, indicating that our conclusions are robust.
>
> We also compute the Intraclass Correlation Coefficient (ICC) between GPT-4o-mini and human judgments. The ICC is 0.89 for Contextual Refusal (high agreement) and 0.57 for Forgetfulness (moderate agreement), with consistent overall trends supporting the reliability of the automatic evaluation.
>
> Table 1: Comparison of different judges on ASRU of Qwen3-VL-8B.
> ||Contextual Refusal $\uparrow$|Forgetfulness $\uparrow$|
> |:-|:-:|:-:|
> |GPT-4o-mini|2.98|3.04|
> |GPT-5.1|2.07|4.6|
> |Claude 4.5|1.3|1.98|
> |Human|2.33|4.37|
> |*ICC*|*0.89*|*0.57*|
> >W2 & Q2: Knowledge-absence direction construction
>
> **R2:** We apologize that the term "absence of knowledge" may be ambiguous. In fact, rather than a "knowledge-free" direction, VL* serves as a reference direction for refusal responses to guide the model toward a "no relevant knowledge and refuse" region on forget set. Empirically, ASRU leads to decreased performance on the forget set, and Figure 5 shows that representations shift toward a concentrated refusal region, indicating that the model effectively learns a meaningful refusal direction.
>
> Additionally, to verify the generalizability of VL*, we apply the VL* constructed on CLEAR to MLLMU (Variant-1). As shown in Table 2, no observable forgetting effect and negligible refusal probability, indicating that VL* does not generalize across datasets and is tied to the specific structure of the forget set.
>
> Table 2: Analysis of VL* on Qwen3-VL-8B.
> ||Forget Cls.|Forget Gen.|Refusal Rollout Rate|
> |:-|:-:|:-:|:-:|
> |Vanilla|55.00|0.60|-|
> |Variant-1|55.83|0.60|0.01|
>
>
> >W3.1 & Q3: Validation of Generalizability
>
> **R3:** To further validate the generalizability of ASRU, we conduct experiments on the LLaVA-v1.5 and CLEAR datasets.
>
> **Evaluating ASRU on LLaVA-v1.5-7B**
> As shown in Table 3, ASRU is effective on LLaVA-v1.5. It achieves notable gains over baselines on forget set, while maintaining model utility on the retain and real sets. ASRU also attains the highest scores in generation quality metrics, demonstrating the effectiveness and generalizability of our method.
>
> Table 3: Unlearning performance for generation task on LLaVA-v1.5-7B (5%Forget).
> |Method|Forget$\downarrow$|Test$\downarrow$|Retain$\uparrow$|Real$\uparrow$|Contextual Refusal$\uparrow$|Forgetfulness$\uparrow$|
> |:-|:-:|:-:|:-:|:-:|:-:|:-:|
> |Vanilla|0.58|0.23|0.49|0.22|-|-|
> |GA_Diff|0.39|0.17|0.35|0.18|0.62|0.92|
> |NPO|0.53|0.22|0.43|0.23|0.04|0.06|
> |MMunlearner|0.42|0.18|0.44|0.23|0.18|0.58|
> |ASRU|0.25|0.14|0.45|0.27|2.12|2.06|
>
> **Evaluating ASRU On CLEAR**
> As seen in Table 4, ASRU also shows excellent performance on CLEAR. Specifically, it achieves gains of 4.79% and 51.61% in classification and generation performance, respectively, on the forget set, while maintaining the utility on the realworld set, further validating the effectiveness and generalizability of our method.
>
> Table 4: Unlearning performance on CLEAR (5% forget) of Qwen3-VL-8B.
> |Method|Forget Cls.$\downarrow$|Forget Gen.$\downarrow$|Realworld Cls.$\uparrow$|
> |:-|:-:|:-:|:-:|
> |Vanilla|78.19|0.42|74.66|
> |GA_Diff|69.14|0.31|67.30|
> |NPO|50.00|0.35|70.02|
> |MMunlearner|75.53|0.32|67.30|
> |ASRU|45.21|0.15|70.03|
> >W3.2: Layer selection
>
> **R4:**  The optimal intervention layer is chosen to maximize the probability of generating refusal responses. Following prior work, we restrict the search to intermediate layers, apply interventions at each candidate layer, simulate rollout in GRPO, and select the layer with the highest refusal probability. As shown in Table 5, layer 17 achieves the highest probability of inducing refusal. Therefore, we select layer 17 as the optimal intervention layer in our experiments.
>
> Table 5: Refusal rollout rates at each layer for Qwen3-VL-4B.
> |Layer|Refusal Rollout Rate$\uparrow$|
> |:-|:-:|
> |16|0.046|
> |17|0.080|
> |18|0.078|
>
> >W3.3: The Choice of Guidance Strength
>
> **R5:** We determine the optimal strength by balancing the preservation of the retain set against the refusal rollout efficacy on the forget set. As shown in Table 6, Qwen3-VL-4B attains the highest Refusal Rollout Rate with minimal effects on the retain set at a steering strength of 3. We follow the same principles for the Qwen3-VL-8B.
>
> Table 6: Impact of steering strength on Qwen3-VL-4B.
> |$\lambda$|Retain $\uparrow$|Refusal Rollout Rate$\uparrow$|
> |:-|:-:|:-:|
> |0.2|0.54|0.01|
> |0.3|0.54|0.01|
> |1.0|0.54|0.01|
> |3.0|0.54|0.05|
>
> If you have further questions, we would be happy to discuss with you.

---

> > ### Author Rebuttal · Reviewer_wYV4 · 2026-04-04
> >
> > Regarding W2&Q2 (knowledge-absence direction construction), the authors clarify that vL* serves as a "reference direction for refusal" rather than a strict knowledge-free direction, and provide evidence that vL* constructed on CLEAR does not transfer to MLLMU (Variant-1), confirming dataset specificity. However, this cross-dataset non-transferability cuts both ways: while it confirms that vL* is tied to the forget set's structure, it equally suggests that the direction may capture dataset-level distributional shift rather than a semantically meaningful knowledge-absence signal, and the rebuttal does not provide the originally requested direct analysis — namely, showing that projecting out vL* from forget-set activations reduces forget-specific information while leaving retain-set activations largely unchanged — which would be the more principled evidence for the claim.
> >
> > I appreciate the authors' efforts and am willing to maintain or slightly raise my score, though I hope the final version clarifies the interpretation of vL* more carefully.

---

> > > ### Author Response · Authors · 2026-04-05
> > >
> > > Thank you for appreciating our efforts and for being willing to raise your score. We address your questions below.
> > >
> > > Following your suggestion, we further directly tested the functional role of $v_{L^\*}$ by projecting out its component from the hidden states at layer $L^\*$=17. Concretely, for a hidden representation $h_{L^\*}(x)$, we apply
> > >
> > > $$h_{L^\*}^{\setminus v}(x)=h_{L^\*}(x)-\frac{\langle h_{L^\*}(x), v_{L^\*}\rangle}{||v_{L^\*}||^2}  v_{L^\*},$$
> > >
> > > The modified hidden state $h_{L^*}^{\setminus v}(x)$ is then fed into the subsequent layers for generation.
> > >
> > > Table 1:projecting out component of $v_{L^*}$ from the hidden states on Qwen3-VL-8B (5% forget), where Steered denotes the steered model after stage 1, and Steered$_{project-out}$ denotes the steered model after projecting out.
> > > | Model | forget | retain |
> > > |:--|:---:|:---:|
> > > | Vanilla | 0.600  | 0.570  |
> > > | Steered | 0.577 | 0.556 |
> > > | Steered$_{project-out}$ | 0.582 | 0.559 |
> > >
> > > As shown in Table 1, we find that this partially reverses the steering effect (forget: 0.577→0.582, retain: 0.556→0.559), showing that $v_{L^*}$ is functionally involved in the induced forgetting/refusal behavior. Compared with the vanilla model, the projection-out model still exhibits a larger drop on the forget set than on the retain set (0.600 → 0.582 vs. 0.570 → 0.559), suggesting that the residual effect remains more forget-oriented than utility-destructive.
> > >
> > > Furthermore, in most unlearning setting, the retain set is highly similar to the forget set in both task format and semantic content, differing mainly in the target identity. For example:
> > > - **Forget set**
> > >   - image_id: 323
> > >   - question: Based on the image, what profession might this person have?
> > >   - answer: This person is likely an Environmental Engineer.
> > >
> > > - **Retain set**
> > >   - image_id: 095
> > >   - question: What is the profession of the individual in the image?
> > >   - answer: The individual is an environmental scientist.
> > >
> > > As can be seen, aside from the difference in the underlying identity (i.e., the person in the image), these two samples are highly similar in both question form and semantic content. Therefore, achieving a perfectly clean separation between the retain set and the forget set is almost impossible under the current unlearning setting. Instead, the practical goal of unlearning is to achieve a favorable trade-off between forgetting on the forget set and utility preservation on the retain set. For this reason, it is difficult to interpret $v_{L^*}$ as being purely semantically meaningful or purely a generic domain-shift vector. Any activation shift applied to suppress the forget set will inevitably affect the retain set to some extent. This is exactly the motivation for our second-stage design: we introduce a boundary set together with a verifiable reward function to further sharpen the boundary between the forget set and the retain set.
> > >
> > > We view this phenomena as consistent with our method design: activation steering provides only a coarse refusal-oriented initialization, while a boundary set and verifiable reward design is introduced to further sharpen the forget-retain boundary.
> > >
> > > We sincerely thank you again for your recognition of our work and for your willingness to raise your score. We will incorporate the additional analysis of $v_{L^*}$ into the revised version.

---

### Official Review · Reviewer_oCQL · 2026-03-11

**Soundness:** 3
**Presentation:** 4
**Significance:** 3
**Originality:** 3
**Overall Recommendation:** 5
**Confidence:** 3

**Summary:**

This paper presents a new 2-stage method based on activation steering and RL for unlearning MLLMs.

**Compliance With Llm Reviewing Policy:**

Affirmed.

**Key Questions For Authors:**

see "Strengths And Weaknesses"

**Limitations:**

yes

**Strengths And Weaknesses:**

This is a very well-written paper that presents a very nice piece of work. The research topic is timely and well-motivated. The unlearning method, outlined in Figure, is technically sound and the results are solid.

I have a few minor questions which might help improve the readability of the paper.

In sec. 3.2.1, need to clarify how to pick the layers where activation steering is applied?

In sec. 3.2.1., will you explain how to curate target knowledge-absence set if we have no knowledge about pre-training data?

In sec. 3, need to justify the use of two stages in ASRU, esp. stage 1. e.g., could we only use RL to optimize Eq. (2)?

How to ensure the quality of the unlearned model, e.g., using a new as the reward of (11) does not seem capture "quality"?

Need to clarify why CR and forgetfulness are good measures of generation quality? Isn't generation quality measured based on the quality/accuracy on the retain dataset?

---

> ### Author Rebuttal · Authors · 2026-03-31
>
> We sincerely appreciate your recognition and insightful review, and address your comments below.
>
> >**Q1: Need to clarify how to pick the layers**
>
> **R1:** The goal of selecting the optimal intervention layer is to maximize the probability of generating refusal responses. Prior work shows that activation steering is generally most effective at intermediate layers. Therefore, our search is primarily restricted to these layers.
>
> Based on this, we perform a layer-wise search over the forget set and simulate the rollout scenario in the second-stage GRPO. Specifically, we apply interventions at each candidate layer and measure the model’s probability of generating a refusal response. The layer maximizing this probability is selected as the optimal intervention layer.
>
> As shown in Table 1, layer 17 achieves the highest probability of inducing refusal and is thus selected as the optimal intervention layer in our experiments.
>
> Table 1: Refusal rollout rates at each layer for Qwen3-VL-4B.
> |Layer|Refusal Rollout Rate$\uparrow$|
> |:-|:-:|
> |16|0.073|
> |17|0.148|
> |18|0.145|
>
> >**Q2: Explain how to curate target knowledge-absence set if we have no knowledge about pre-training data**
>
> **R2:** In our method, $D_{\text{target}}$ serves as a reference set for the refusal direction, without requiring strict verification of its presence in the pre-training data. Even if the pre-training data is inaccessible, $D_{\text{target}}$ can still be constructed based on task-level principles. Specifically, we expect it to satisfy two criteria:
>
> (1) Its input format should closely match the forget queries to avoid learning irrelevant distributional differences；
>
> (2) The model should produce contextually appropriate refusals for these inputs, providing a reference for the “knowledge missing / should refuse” state.
>
> Based on these principles, $D_{\text{target}}$ can be built using fictitious or synthetic identities or unanswerable privacy-related inputs. These samples need not be confirmed as “seen” during pretraining; rather, they form a semantically appropriate reference set for refusal at the task level.
>
> >**Q3: Need to justify the use of two stages in ASRU, esp. stage 1. e.g., could we only use RL to optimize Eq. (2)?**
>
> **R3:** Stage 1 uses activation steering (via updating a single projection matrix) to induce controllable refusal behavior as a lightweight initialization for Stage 2. Without it, the model is unable to generate refusal trajectories during the rollout of the forget set in Stage 2. Thus, Stage 1 is critical in establishing a basic refusal prototype, while Stage 2 refines this prototype to achieve a more granular distinction between forget and retain regions. These two stages are complementary and cannot be replaced by a single RL-based optimization. The results from our ablation study on Table 2 of the paper demonstrate that removing Stage 1 significantly impairs the model’s ability to refuse appropriately, thereby negatively affecting unlearning performance.
>
>
> >**Q4: How to ensure the quality of the unlearned model, e.g., using a new as the reward of (11) does not seem capture "quality"?**
>
> **R4:** The verifiable reward function is designed to encourage the model to exhibit context-aware, dynamic refusal behaviors on the forget set while generating informative responses on the retain set. In this way, reinforcement learning guides the model to learn a fine-grained forgetting boundary.
>
> To enable refusal-aware reward design, we define a set of regular expression patterns in Appendix C.1 to match natural language expressions that convey epistemic uncertainty (e.g., “I don’t know,” “I am not sure”). These patterns determine whether a model output $y$ constitutes a valid refusal.
>
> >**Q5: Need to clarify why CR and forgetfulness are good measures of generation quality? Isn't generation quality measured based on the quality/accuracy on the retain dataset?**
>
> **R5:** Existing methods typically claim success by measuring the deviation between the model outputs on the forget set and the ground truth. However, unlearned model often exhibit hallucinatory or rigid responses as shown in Figure 1 of the paper, which can pose significant risks in high-stakes scenarios (e.g., medical settings).
>
> *This raises a key question: can an unlearned model accurately express its knowledge gaps in a dynamic, context-aware, and coherent manner?*
>
> Furthermore, an ideal unlearned model should forget only the targeted knowledge without affecting neighboring or semantically similar knowledge.
>
> Therefore, we propose CR and forgetfulness to evaluate whether the unlearned model produces contextually refusal responses on the forget set, while performance on the retain set primarily assesses the preservation of model utility.
>
> We will incorporate all the above results and analyses into the revised
> paper.

---

> > ### Author Rebuttal · Reviewer_oCQL · 2026-04-03
> >
> > The responses are good. I will keep my rating.

---

> > > ### Author Response · Authors · 2026-04-04
> > >
> > > We sincerely thank you for the positive feedback on our work and for the constructive suggestions. We especially appreciate your recognition of the **motivation**, **technical design**, and **empirical evaluation** of our method.
> > >
> > > Our work is motivated by a practical limitation of existing unlearning methods for MLLMs: unlearned models often produce hallucinated or overly rigid responses, which can pose substantial risks in high-stakes scenarios such as medical applications. *We argue that an ideal unlearned model should not merely suppress target knowledge, but should also be able to express its knowledge gaps accurately in a dynamic, context-aware, and coherent manner.*
> > >
> > > To address this issue, we propose ASRU, a two-stage framework that first uses activation steering to induce initial refusal behavior by updating a single projection matrix, and then introduces a boundary set and verifiable rewards to learn a finer-grained forget–retain boundary. In this way, ASRU not only removes target knowledge, but also encourages more natural and reliable responses after unlearning while preserving model utility.
> > >
> > > Thank you again for your recognition of our work. In the revised version, we will incorporate the above experimental results and analyses.

---

### Official Review · Reviewer_dYwq · 2026-03-12

**Soundness:** 3
**Presentation:** 3
**Significance:** 3
**Originality:** 3
**Overall Recommendation:** 4
**Confidence:** 2

**Summary:**

This paper studies multimodal unlearning for MLLMs and argues that existing methods usually evaluate forgetting mainly by output deviation, while paying too little attention to whether the post-unlearning responses are still natural, coherent, and context-aware. To address this, the authors propose ASRU, a two-stage framework that first uses activation steering to induce a basic refusal behavior by updating a single down-projection matrix, and then applies GRPO-based reinforcement learning with a customized reward and a boundary set sampled from retained data to learn a finer-grained “refuse when necessary, answer when appropriate” boundary.

**Compliance With Llm Reviewing Policy:**

Affirmed.

**Key Questions For Authors:**

1. The generation-quality part is interesting, but since it is evaluated by ChatGPT-4o Mini, how confident are you that this score really matches human judgment and is not overly sensitive to the evaluator or prompt design?

2. In the ablation, removing the boundary set gives even stronger forgetting, but utility drops a lot. How sensitive is the full method to how that boundary set is constructed, and would it still work well if those boundary samples were noisier or less similar to the forget set?

3. The experiments are mainly on image-text privacy-style unlearning with Qwen-3-VL models. Do you expect the same activation-steering-plus-GRPO recipe to transfer cleanly to video or audio settings, or do you think the refusal boundary becomes much harder to learn there?

**Limitations:**

yes

**Strengths And Weaknesses:**

The paper has a clear and timely motivation, because it points out a real weakness in current multimodal unlearning work: a model can “forget” in metric space yet still respond in a hallucinated, awkward, or overly rigid way, which is especially problematic in privacy-sensitive settings. I also think the method is reasonably well structured: the activation-steering stage provides a meaningful initialization for refusal behavior, and the GRPO stage then refines the forget-retain boundary instead of relying on crude suppression alone.

The main weakness is that the evaluation story still depends quite a lot on the paper’s own framing of generation quality, and those scores are judged by ChatGPT-4o Mini, so some readers may worry about subjectivity, evaluator bias, or metric sensitivity compared with more standardized human evaluation. I also think the claimed novelty is somewhat mixed: activation steering and RL-based alignment are both established ingredients, so the contribution is more in how they are combined for multimodal unlearning than in introducing a fundamentally new learning principle.

---

> ### Author Rebuttal · Authors · 2026-03-31
>
> We decompose your concerns into two points and address them below
> >**W1 & Q1: The robustness of evaluation using GPT-4o-mini.**
>
> **R1:**  We further re-evaluate the responses using GPT-5.1 and Claude 4.5. As shown in Table 1, although the scores vary across judges, the relative ranking of methods remains consistent, and ASRU achieves the best performance under all evaluators, demonstrating the robustness of our conclusions.
>
> Table 1: Generation quality evaluation by different LLM judges.
> ||Contextual Refusal $\uparrow$|||Forgetfulness $\uparrow$|||
> |:-|:-:|:-:|:-:|:-:|:-:|:-:|
> |Model|GPT-4o-mini|GPT-5.1|Claude 4.5|GPT-4o-mini|GPT-5.1|Claude 4.5|
> |GA|0.32|0.42|0.28|0.34|0.52|0.70|
> |GA diff|0.68|0.38|0.08|0.72|0.58|0.60|
> |NPO|0.56|0.56|0.44|0.58|0.74|0.70|
> |KL Min|0.32|0.38|0.28|0.36|0.48|0.48|
> |MMunlearner|0.20|0.36|0.20|0.24|0.44|0.54|
> |ASRU (ours)|2.98|2.07|1.30|3.04|4.60|1.98|
>
>
> To further verify the consistency between GPT-4o-mini and human judgments, we invite 3 annotators to evaluate the samples and compute the Intraclass Correlation Coefficient (ICC),a standard metric for inter-rater agreement(0 to 1,higher is better). As shown in Table 2, the ICC for CR is 0.89, indicating high agreement, while the ICC for Forgetfulness is 0.57, reflecting moderate agreement. And the overall trends remain consistent, supporting the reliability of the automatic evaluation.
>
> Table 2: Agreement between GPT-4o-mini evaluation and human judgments.
> |Metrics|GPT-4o-mini|Human|ICC|
> |-|:-:|:-:|:-:|
> |Contextual Refusal|2.98|2.33|0.89|
> |Forgetfulness|3.04|4.37|0.57|
>
> >**W2: The claimed novelty is somewhat mixed.**
>
> **R2:** We apologize for any lack of clarity in presenting the novelty of our work. Our intention is to propose a structured and targeted framework for multimodal unlearning. The main contributions are summarized as follows:
>
> **Problem formulation:**
> Existing multimodal unlearning methods primarily focus on output deviation. We argue that the quality of post-unlearning responses, whether the model produces natural and context-aware refusals, is also a critical yet underexplored dimension.
>
> **Method design:**
> Our method is not a simple pipeline of activation steering and RL, but a structured decomposition where each stage serves a distinct role:
>
> (i) Stage 1 uses activation steering (via updating a single projection matrix) to induce controllable refusal behavior as a initialization;
>
> (ii) Stage 2 introduces a boundary set and verifiable reward design to learn a finer-grained forget–retain boundary.
> The key idea is to first establish a refusal prototype, and then refine the decision boundary.
>
> **Adaptation to multimodal unlearning:**
>
> Our design is tailored to the challenge of multimodal unlearning, i.e., removing target knowledge while preserving closely related information, and enabling context-appropriate refusals rather than hallucinated or rigid outputs. In particular, the boundary set construction and reward design are specifically developed to improve selective refusal and generation quality.
>
> In summary, our contribution is better characterized as a novel framework design and empirical study for multimodal unlearning. We will revise the manuscript to clarify this positioning and align the novelty claims accordingly.
>
>
> >**Q2: Sensitivity of the boundary set construction**
>
> **R3:** To evaluate the sensitivity of the boundary set, we replace a portion (50% and 100%) of the boundary set with noise samples unrelated to our task.
> As shown in Table 3, as more noise data is injected, the model’s performance on the retain and real sets deteriorates progressively, indicating that, to facilitate the model’s learning of a more fine-grained refusal boundary, the construction of the boundary set should consist of samples that are similar to or neighbors of the forget set.
>
> Table 3: Sensitivity of the boundary set construction on Qwen3-VL-8B (5% forget).
> |Method|forget$\downarrow$|retain$\uparrow$|real$\uparrow$|
> |:-|:-:|:-:|:-:|
> |Vanilla|0.60|0.57|0.43|
> |ASRU|0.40|0.54|0.40|
> |-w 50% noise|0.48|0.50|0.36|
> |-w 100% noise|0.29|0.30|0.24|
>
> >**Q3: Transfer to video or audio settings.**
>
> **R4:** We believe that, in principle, ASRU could potentially be extended to video or audio scenarios, since its core does not rely on the specific input modality of static image–text pairs, but rather on two more general steps.
>
> At the same time, we acknowledge that such a transfer may not be straightforward. Compared with images, video and audio typically involve stronger temporal dependencies and more complex cross-modal couplings. Knowledge to be forgotten may also be distributed across multiple frames, long-range acoustic patterns, or inter-segment semantic interactions. Consequently, in these scenarios, learning the refusal boundary may become more challenging. Therefore, we regard the extension to video/audio as an important direction for future research.
>
> We will incorporate all the above results and analyses into the revised paper.

---

> > ### Author Rebuttal · Reviewer_dYwq · 2026-04-05
> >
> > My comments have been addressed. I will maintain my positive score.

---

> > > ### Author Response · Authors · 2026-04-07
> > >
> > > We sinerely thank you for the positive feedback on our work and recognition of our work. We appreciate your acknowledgment of **the clear and timely motivation**, **the sound methodological design**, and **the well-structured presentation of the paper**.
> > >
> > > In response to your concerns, we have provided additional evidence on the robustness of the judge model, further clarified our contributions, and analyzed the sensitivity of our method to the construction of the boundary set. We also outline promising directions for future work.
> > >
> > > We sincerely thank you and appreciate your valuable suggestions. We will incorporate these clarifications and analyses into the revised version.

---

### Official Review · Reviewer_WMF5 · 2026-03-13

**Soundness:** 3
**Presentation:** 3
**Significance:** 3
**Originality:** 2
**Overall Recommendation:** 5
**Confidence:** 3

**Summary:**

The paper proposes ASRU, a two-stage framework for multimodal unlearning in multimodal models. In stage 1, it constructs a knowledge-absence direction by contrasting hidden activations from the forget set with activations from a target knowledge-absence set, then steers the model toward refusal behavior by updating a single down-projection layer. In stage 2, it applies GRPO on forget examples plus a similarity-based boundary subset from the retain set, using a custom reward to encourage more selective refusal rather patterns that may suffer during stage 1.

**Compliance With Llm Reviewing Policy:**

Affirmed.

**Final Justification:**

This paper proposes a new framework for multimodal unlearning in multimodal models. This topic is well discovered in textual domain, however is still underrepresented for the multimodal models. The conducted experiments in the original paper, and additional results from the rebuttal stage make the paper an valuable piece of research in this domain. I suppose that the paper can be accepted for the conference.

**Key Questions For Authors:**

1) The evaluation of contextual refusal and leakage relies on GPT-4o mini as a judge. Did you test the robustness of these conclusions with other judge models?
2) To what extent does ASRU achieve true unlearning rather than simply learning to refuse answers in the benchmark format? Did you test paraphrases, adversarial prompts, or multi-turn extraction settings?
3) In textual unlearning, an important property is locality: forgetting one fact should not erase nearby related facts. Did you evaluate an analogous notion of locality in the multimodal setting?

**Limitations:**

I think that the main limitations are as follows:
1) Judge-based evaluation is limited.
2) True unlearning vs refusal is not fully disentangled. Like how sensitive is the model to augmentation of the prompt or adversarial attacks.
3) Locality of forgetting is underexplored.

**Strengths And Weaknesses:**

1) The paper is technically sound and addresses the important problem of unlearning in multimodal models. This is a highly relevant research direction, since in most cases these challenges have been studied primarily in the text-only domain.
2) The paper is well written and clearly structured, and its claims are generally supported by the experiments presented.
3) The problem addressed is significant and important. As mentioned above, unlearning has been explored mostly in the textual domain, while multimodal connections are also crucial. It is also valuable that the paper analyzes refusal behavior in the models.
4) The work does not propose an entirely new method, but the modifications and updates are still quite original.

The main weaknesses are as follows:
1) The use of GPT-4o mini as a judge seems somewhat weak. It is an older model, so it would be helpful to know whether the authors considered using other models for evaluation.
2) I understand that there are not many benchmarks for multimodal forgetting. However, an important issue is that, in the textual domain, unlearning is also expected to preserve closely related knowledge. For example, forgetting that the Eiffel Tower is in France should not affect the fact that Paris is the capital of France. This kind of locality can also be considered in multimodal models.

---

> ### Author Rebuttal · Authors · 2026-03-31
>
> >**W1 & Q1:** The use of GPT-4o mini as a judge seems somewhat weak.
>
> **R1:** Thank you for this important suggestion. To further validate the reliability of our experimental results, we additionally re-evaluate the responses using **GPT-5.1** and **Claude 4.5** as independent judge models.
>
> As shown in Table 1, although the scores vary across judges, the relative ranking of methods remains consistent, and ASRU achieves the best performance under all evaluators, demonstrating the robustness of our conclusions.
>
> Table 1: Generation quality evaluation by different LLM judges on Qwen3-VL-8B (5% Forget).
> ||Contextual Refusal $\uparrow$|||Forgetfulness $\uparrow$|||
> |:-|:-:|:-:|:-:|:-:|:-:|:-:|
> |**Model**|**GPT-4o-mini**|**GPT-5.1**|**Claude 4.5**|**GPT-4o-mini**|**GPT-5.1**|**Claude 4.5**|
> |GA|0.32|0.42|0.28|0.34|0.52|0.70|
> |GA_diff|0.68|0.38|0.08|0.72|0.58|0.60|
> |NPO|0.56|0.56|0.44|0.58|0.74|0.70|
> |KL_Min|0.32|0.38|0.28|0.36|0.48|0.48|
> |MMunlearner|0.20|0.36|0.20|0.24|0.44|0.54|
> |ASRU (ours)|2.98|2.07|1.30|3.04|4.60|1.98|
>
> To further verify the consistency between GPT-4o-mini and human judgments, we invite 3 annotators to evaluate the samples and compute the Intraclass Correlation Coefficient (ICC) , a standard metric for inter-rater agreement (values >0.75 indicate high agreement, 0.4–0.75 moderate agreement, and <0.4 low agreement). As shown in Table 2, the ICC for CR is 0.89, indicating high agreement, while the ICC for Forgetfulness is 0.57, reflecting moderate agreement. And the overall trends remain consistent, supporting the reliability of the automatic evaluation.
>
>
> Table 2: Agreement between GPT-4o-mini evaluation and human judgments.
> |Metrics|GPT-4o-mini|Human|ICC|
> |-|:-:|:-:|:-:|
> |**Contextual Refusal**|2.98|2.33|0.89|
> |**Forgetfulness**|3.04|4.37|0.57
>
>
> >**W2 & Q3:** Locality of forgetting
>
> **R2:** Thank you for raising this important point. In the textual domain, locality typically requires that the unlearning process should be as precise as possible, affecting only the target knowledge itself without unnecessarily disturbing neighboring or semantically related knowledge [1]. We believe that this requirement is equally important in the multimodal setting.
>
> In our experimental setup, the **retain set** can be viewed as a form of neighbor perturbation, and its evaluation already reflects locality to some extent. Specifically, the retain set is highly similar to the forget set in both task format and semantic content, differing mainly in the target identity. For example:
>
> - **Forget set**
>   - image_id: 323
>   - question: Based on the image, what profession might this person have?
>   - answer: This person is likely an Environmental Engineer.
>
> - **Retain set**
>   - image_id: 095
>   - question: What is the profession of the individual in the image?
>   - answer: The individual is an environmental scientist.
>
> As can be seen, apart from the fact that the underlying person is different, these two samples are highly similar in both question form and semantic content.
>
> [1] Zhuoran jin, Pengfei Cao, et al. 2024. RWKU: Benchmarking Real-World Knowledge Unlearning for Large Language Models. Advances in Neural Information Processing Systems (NeurIPS 2024).
>
>
> >**Q2:** Test on paraphrases, adversarial prompts, or multi-turn extraction settings.
>
> **R2:** Thanks for your insightful question. We design three types of prompt variants to systematically evaluate the robustness of the refusal boundary learned by ASRU from different perspectives:
>
> - Random Prefix: We prepend semantically neutral prefixes (e.g., "The is a piece of news.") to the original queries to test robustness against lightweight surface perturbations.
>
> - Paraphrase: We use GPT-5.1 to generate three semantically equivalent but lexically diverse paraphrases for each query, evaluating generalization to natural language variation.
>
> - Jailbreak-style Prompt: We prepend adversarial instructions (e.g., "You’re an AI with access to vast knowledge ...") to explicitly encourage the model to bypass the learned refusal boundary.
>
> As shown in Table 3, for both generation and classification tasks, the performance of each model on the forget set remains stable or even slightly decreases under different prompt variants. This consistent behavior indicates that ASRU does not rely on fixed template matching but instead learns a more generalizable refusal boundary.
>
> Table 3: Unlearning performance under different prompt variants (5% forget).
> ||**Qwen3-VL-4B**||
> |:-|:-:|:-:|
> |**Prompt**|**Gen:RougeScore**|**Cls:ACC(%)**|
> |**Original**|0.399|31.20|
> |**Random Prefix**|0.364|29.60|
> |**Paraphrase**|0.392|30.40|
> |**Jailbreak Prompt**|0.406|31.20|
> ||**Qwen3-VL-8B**||
> |**Original**|0.398|30.83|
> |**Random Prefix**|0.221|16.00|
> |**Paraphrase**|0.206|20.00|
> |**Jailbreak Prompt** |0.245|20.00|
>
> We will incorporate all the above results and analyses into the revised paper.

---

> > ### Author Rebuttal · Reviewer_WMF5 · 2026-04-05
> >
> > I thank the authors for the response. I have no more following questions. I suppose I will increase my score to a more confident level.

---

> > > ### Author Response · Authors · 2026-04-07
> > >
> > > Thank you very much for your positive feedback and recognition of our work. We are glad that you recognized the **technical soundness** of our paper and **the value of our analysis of refusal behavior**. We also appreciate that you found the paper **well written and generally well supported by experiments**.
> > >
> > > In response to your concerns, we have provided additional evidence on the robustness of the judge model, further elaborated on the notion of locality, and evaluated the robustness of our method under adversarial settings.
> > >
> > > We are grateful for your thoughtful suggestions and will incorporate these clarifications and analyses into the revised version.

---

### Decision · Program_Chairs · 2026-04-30

**Decision:**

Accept (regular)

**Comment:**

The paper proposes ASRU, a two-stage framework for multimodal unlearning in multimodal large language models. In the first stage, it constructs a knowledge-absence direction by contrasting hidden activations between the forget set and a target knowledge-absence set, and steers the model toward refusal behavior by updating a single projection layer. In the second stage, it refines this behavior using GRPO-based reinforcement learning with a custom reward and a boundary set drawn from the retain data, enabling the model to selectively refuse harmful queries while maintaining coherent and context-aware responses. Across reviews, the method is positioned as addressing a key limitation in prior work: balancing effective forgetting with natural, high-quality generation.

Strengths
- Addresses an important and timely problem: multimodal unlearning, extending beyond the largely text-focused literature
- Clear motivation highlighting the gap between forgetting effectiveness and generation quality
- Technically sound and well-structured two-stage framework combining activation steering and RL
- Efficient design, including lightweight parameter updates and principled initialization
- Strong empirical results with generally well-supported claims and useful ablations
- Introduces boundary set construction to better balance forgetting and retention
- Well-written paper with clear presentation and illustrative examples

Weaknesses
- Heavy reliance on GPT-4o Mini as an evaluator raises concerns about metric reliability, bias, and lack of human validation
- Limited evaluation scope (single benchmark, model family, and domain), weakening generalization claims
- Novelty is somewhat incremental, as it combines existing techniques rather than introducing fundamentally new methods
- Insufficient analysis of whether true unlearning is achieved versus learned refusal behavior
- Lack of robustness checks (e.g., paraphrases, adversarial prompts, multi-turn settings)
- Underexplored locality of unlearning (impact on related knowledge)
- Some methodological clarity issues (e.g., layer selection, dataset construction, reward design)

Almost all concerns have been addressed by the authors during the rebuttal period. I am recommending accepting this paper.